# Coenzyme Q_10_ Supplementation and Its Impact on Exercise and Sport Performance in Humans: A Recovery or a Performance-Enhancing Molecule?

**DOI:** 10.3390/nu14091811

**Published:** 2022-04-26

**Authors:** Franchek Drobnic, Mª Antonia Lizarraga, Alberto Caballero-García, Alfredo Cordova

**Affiliations:** 1Medical Services Shanghai Shenhua FC, Shanghai 201315, China; 2Medical Services FC Barcelona, 08014 Barcelona, Spain; mlizarraga@ub.edu; 3Department of Anatomy and Radiology, Faculty of Health Sciences, GIR: “Physical Exercise and Aging”, Campus Universitario “Los Pajaritos”, University of Valladolid, 42004 Soria, Spain; alberto.caballero@uva.es; 4Department of Biochemistry, Molecular Biology and Physiology, Faculty of Health Sciences, GIR: “Physical Exercise and Aging”, Campus Universitario “Los Pajaritos”, University of Valladolid, 42004 Soria, Spain; a.cordova@uva.es

**Keywords:** Coenzyme Q_10_, ubiquinone, ubiquinol, mitoquinone, nutritional supplement

## Abstract

Evidence exists to suggest that ROS induce muscular injury with a subsequent decrease in physical performance. Supplementation with certain antioxidants is important for physically active individuals to hasten recovery from fatigue and to prevent exercise damage. The use of nutritional supplements associated with exercise, with the aim of improving health, optimizing training or improving sports performance, is a scientific concern that not only drives many research projects but also generates great expectations in the field of their application in pathology. Since its discovery in the 1970s, coenzyme Q_10_ (CoQ_10_) has been one of the most controversial molecules. The interest in determining its true value as a bioenergetic supplement in muscle contraction, antioxidant or in the inflammatory process as a muscle protector in relation to exercise has been studied at different population levels of age, level of physical fitness or sporting aptitude, using different methodologies of effort and with the contribution of data corresponding to very diverse variables. Overall, in the papers reviewed, although the data are inconclusive, they suggest that CoQ_10_ supplementation may be an interesting molecule in health or disease in individuals without a pathological deficiency and when used for optimising exercise performance. Considering the results observed in the literature, and as a conclusion of this systematic review, we could say that it is an interesting molecule in sports performance. However, clear approaches should be considered when conducting future research.

## 1. Introduction

The use of performance-enhancing substances is current but not new to mankind. Substances to increase performance, reduce fatigue, facilitate recovery and even modify the will have been present and used since ancient times in the form of stimulants, defatigating or anabolic agents [1]. Therefore, a performance improvement is the combination of increased efficiency and effectiveness and is conceived as the achievement of the desired goal. In the field of sport, an improvement in performance can be considered as achieving the objectives of the chosen sport because to the skills acquired or the elements provided to improve it [2].

Associated with this training there is a wide range of support where nutritional supplementation has its place, either directly or indirectly. We can consider a nutritional supplement to be that extra contribution to the usual diet in order to complement it, to meet the requirements produced by an excess of us or because the characteristics of the substance provided offer qualities that improve or facilitate the sporting activity or work. The objective of its use, whether that substance improves health or the physical or sporting action itself, is to achieve optimum performance within the sphere of health. In a more practical and concrete way, the definition of supplement could be defined as “a substance that is purposefully ingested in addition to the habitually-consumed diet with the aim of achieving a specific health and/or performance benefit” [3].

Some supplements certainly have strong evidence bases to reflect a direct impact on athletic performance through the augmentation of various rate-limiting processes. However, other supplements may have an indirect impact on performance via their ability to support the training process, through their influence on factors such as inflammatory modulation, oxidative stress and signaling pathways for adaptation or their ability to support repetitive performance by restoring homeostasis between two exercise bouts. [4].

The reactive oxygen species (ROS) are a family of molecules that are continuously generated and consumed in all living organisms as a consequence of aerobic life [5]. The biological impact of ROS depends not only on their quantities but also on their chemical nature, (sub)cellular and tissue location and the rates of their formation and degradation. ROS represent key modulators of cellular physiology triggering adaptive responses when produced in limited amounts but are detrimental when they are produced in excess, leading to oxidative stress and cellular dysfunction [6]. Enhanced metabolic rate together with a rise in temperature and a decrease in cellular pH could accelerate ROS production [7], particularly in mitochondria, where constitutively 1–2% of oxygen is converted into superoxide anion [8]. As a result, the exercising muscle produces much larger amounts of ROS compared to the muscle at rest [9].

Skeletal muscle has a high capacity to adapt to various demands. Interestingly, the functions of skeletal muscle and their neighboring cells are altered by oxidative stress modulation [10]. Macrophages are a major source of reactive oxygen species (ROS) and reactive nitrogen species (RNS) in inflammation [11]. Inflammatory macrophages release glutathionylated peroxiredoxin-2, which acts as a “danger signal” to trigger the production of tumour necrosis factor alpha [12]. ROS exert major biological effects as purely deleterious molecules causing damage to DNA, proteins and lipids, notably during exercise [13]. Evidence exists to suggest that ROS induce muscular injury with a subsequent decrease in physical performance [12,13]. Supplementation with certain antioxidants is important for physically active individuals to hasten recovery from fatigue and prevent exercise damage [14].

There are fundamental molecules in the organism that we have been studying for years as nutritional supplementation and which, for various reasons, have not yet defined their true role in the possible improvement of sports performance such as CoQ_10_. The CoQ_10_ molecule is a fat-soluble, vitamin-like, ubiquitous compound, with a key role in cellular bioenergetics, because act as a cofactor in the mitochondrial respiratory chain to supply energy for cells [15,16].

CoQ_10_ participates in redox reactions within the electron transport chain at the mitochondrial level facilitating the production of adenosine triphosphate (ATP) [15] to a mobile redox agent shuttling electrons and protons in the electron transport chain. The CoQ_10_ have more functions that its role in the mitochondria. These including lipophilic antioxidant effect that protects the DNA [17], the phospholipids and the mitochondrial membrane proteins against the lipid peroxidation [18,19]. CoQ_10_ also help to regenerates vitamins C and E and reduces inflammatory markers [15,20].

CoQ_10_ acts as an antioxidant in both the mitochondria and lipid membranes by scavenging reactive oxygen species (ROS), either directly or in conjunction with a-tocopherol [15,21,22]. This antioxidant activity appears only with the reduced form (ubiquinol). The oxidized form (ubiquinone) is readily reduced to ubiquinol enzymically after dietary uptake [23].

The research on the impact of Ubiquinone, and its reduced form Ubiquinol, as sup-plementation in exercise in humans began in the late 1980s. More recently labdeveloped Mitoquinone [24,25,26,27]. Unfortunately, there is no consistency in the studies due to the diversity of functions of CoQ_10_ in the body that can be applied to the improvement of physical performance. There are a wide dispersion of study models allowing the coexistence of different dose, patterns of administration, vehicle or diet to enhance bioavailability, subjects considered present diverse physical capacities or sports experience, are of various ages, and undertake different types of stressing exercise if it is done, etc.

Given the methodological diversity, it is difficult to consider uniformity of studies, which could increase the scientific evidence. Greater uniformity can be achieved if results are evaluated without considering methodologies. For this reason, it is not possible to elaborate an ideal review to elucidate the effect of Q_10_ on sports performance. Nor would it be feasible in studies on its use in clinical therapeutics, ageing and disease [27,28,29,30,31]. However, there are previous reviews that provide interesting data on the subject, some of them recent [32], and some of them extraordinary to understand the basis for the usefulness of CoQ_10_ as an ergogenic substance and its limitations. Previously, Malm et al. [33] and Sarmiento et al. [34] conducted studies of great interest for the same purpose.

The current work that we present reviews, in a holistic and at the same time pragmatic way, the usefulness of CoQ_10_ in sports performance. The main objective of this review was to critically evaluate the effectiveness of CoQ_10_ supplementation on physical performance. Also, analyse its effects as inflammatory and oxidative factors, in a physically active population and specially in athletes.

## 2. Methods

### 2.1. Search Strategy

The literature search was conducted according to the guidelines for meta-analysis of systematic reviews (PRISMA). [35,36,37]. The studies were evaluated according to variables related to the type and size of the population studied, the administration regimen of the product, its bioavailability, the existence or not of a placebo and/or control group, the tests used to assess performance improvement and the appropriateness of the method used. Of course, all in relation to our objective, when Ubiquinone [38,39,40,41,42,43,44,45,46,47,48,49,50,51,52,53,54,55,56,57,58,59,60,61,62,63,64,65,66,67,68,69,70,71,72,73,74,75,76,77,78,79], Ubiquinol [80,81,82,83,84,85,86,87,88,89,90] or Mitoquinone [80,91,92,93] was administered.

A structured search was carried out in the databases SCOPUS, Medline (PubMed), and Web of Science (WOS), which includes other databases such as BCI, BIOSIS, CCC, DIIDW, INSPEC, KJD, MEDLINE, RSCI, SCIELO, all of which are high-quality databases which guarantee good bibliographic support. The search was performed from the first available date, according to the characteristics of each database until February 2022 (time of last update).

### 2.2. Selection of Articles: Inclusion and Exclusion Criteria

For the articles obtained in the search, the following inclusion criteria were applied to select studies: articles depicting a well-designed experiment that included the ingestion of a dose of CoQ_10_, or a CoQ_10_ product, before and/or during exercise in humans. It was considered appropriate to use the descriptors “Ubiquinone”, “Coenzyme Q_10_”, “Ubiquinol”, “Q_10_”, “Mitoquinone” combined with “Exercise”, “Athlete”, “Performance”, Sport” and “Physical Exertion”. The limit “Humans” was applied.

The suitability of the articles was determined using the GRADE [94] and the level of evidence criterion [95]. All analysed articles demonstrated a moderate or high scientific quality, and those with a degree of evidence that can be classified from 2 to 2++ were selected. With the same concept as the Sarmiento et al. review [34], no restrictions on the participants’ gender, age or fitness level were established initially. Inclusion criteria were then applied, which were: randomised, double-blind, controlled, parallel design studies assessing plasma CoQ_10_. Likewise, the studies would be about the assessment of the inflammatory and oxidative effects, and aspects related to muscle injury or its impact on physical performance.

On the other hand, and because of the objective of the substance administration, in studies where the evaluation of its impact on physical and sporting performance had to be assessed, it should be carried out with athletes or subjects who were trained and comfortable with performing intense exercise. Finally, the measurement of the CoQ_10_ in muscle-by-muscle biopsy was also considered an important and necessary aspect, as it is the target site of the functionality of the molecule under study. The “Full search strategy” is presented below.

Study quality was assessed according to the PEDro scale (https://pedro.org.au/english/resources/PEDro-scale/) (accessed on 4 September 2021). The PEDro scale has strong reliability and validity [96,97]. The scale consists of a list of 11 items; for each individual criterion of scientific methodology, studies receive a score of 1 when the criterion is clearly met or 0 when that criterion is not adequately met. The first item (eligibility criteria) does not receive a score because it is related to external validity and, therefore, does not reflect the quality dimensions assessed by the PEDro scale. The PEDro score for each study is shown in Table 1.

Thus, the total scores range from 0 to 10. The criteria included in the Scale are:Eligibility criteria were specified (no score).Subjects were randomly allocated to groups.Allocation was concealed.The groups were similar at baseline regarding the most important prognostic indicators.There was blinding of all subjects.There was blinding of all therapists who administered the therapy.There was blinding of all assessors who measured at least one key outcome.Measures of at least one key outcome were obtained from more than 85% of the subjects initially allocated to groups.All subjects for whom outcome measures were available received the treatment or control condition as allocated or, where this was not the case, data for at least one key outcome was analysed by “intention to treat”.The results of between-group statistical comparisons are reported for at least one key outcome.The study provides both point measures and measures of variability for at least one key outcome.


## 3. Results

Of the articles viewed, 59 were related to the select descriptors, and only 9 met all the inclusion/exclusion criteria (Figure 1). Four, out of the 59, were duplicated in any sense, Drobnic [47] and Paredes-Fuentes [98], Tauler [69] and Ferrer [70], the two studies by Ciocoi-Pop [44], and Diaz-Castro [83,84]. The data for the studies that meet all the proposed inclusion criteria are listed together with those that only lack a muscle biopsy in Table 2 and Table 3. The remaining studies are listed in Table 4, Table 5, Table 6, Table 7, Table 8 and Table 9. Only Svensson’s 1999 study [68] fulfilled all criteria. These authors suplemented the subjects with 120 mg/day of Ubiquinone. In the results they observed an increase in plasma CoQ_10_ levels after administration. However, it did not increase in muscle. Also, was not observed improvement in the oxidative pattern after physical assessment tests. Of the other eight studies selected, four are with Ubiquinone and the other four with Ubiquinol. All observed elevation of baseline plasma levels after treatment. Of the nine studies, only six assessed the impact on physical performance, none on sports performance. Two, Suzuki [90] and Kon [86], found a positive impact, while Svensson [68] Kizaki [85], Braun [41] and Weston [72] did not. Interestingly, there is a positive response on the inflammatory [83,89] and antioxidant pattern to the stress [41,56,86,89], bone remodelling [85] and the metabolic [57] response.

The inclusion criteria administered in the review are indeed strict. It is true that some of them could be omitted, e.g., that the study can be cross-over, that studies with a control group and not only placebo are included, or that it is not determined whether or not the tested molecule increases in tissues. If the discussion focuses not only on the objective but considers a correct methodological structure for assessing the effect on performance, then there would be various deselected studies that certainly deserve to be considered. For this reason, Table 4, Table 5, Table 6, Table 7, Table 8 and Table 9 include the rest of the studies screened and some of them are discussed.

In the tables the dosage administered is indicated in mg per day (mg/d) and in those studies where it was offered together with vitamin C, E or another substance, it is indicated with a symbol (+) next to the molecule administered. The number of subjects included in the study is shown in the “Total/CoQ_10_” column, where those who took the molecule in the study are indicated in reference to the total number of subjects in the study. The crossover studies show the same number on both sides of the slash bar. The quality of physical fitness is shown in the column “Type of subjects”. To homogenize the groups, although there is no consistent definition in the study methodologies in this respect, it has been considered in a somewhat arbitrary way but determined by the existing indications where available, the quality of the tests performed and the data of the subjects, in five different groups “High level”, “Well trained subjects”, “Moderately trained subjects”, “Physically active subjects”, “No active subjects”, “Patients”. The difference between the groups “Moderately trained subjects” and “Physically active subjects” depends on the quality of the indications in the studies. The first ones used to practice sport or have a regular weekly training schedule. However, the second are only, as the concept indicates, “physically active subjects” and they cannot be considered sedentary or specifically, “no active”. The physical tests are indicated in the tables when it was performed. From the 55 studies, in 22 were performed a graded maximal exercise test, and in 8, specific anaerobic tests. Other exercise stress methods, either sport-specific, with field or swimming pool tests, or custom-made laboratory tests, categorized as maximal, sub-maximal, evaluating energy metabolism, degree of fatigue, or the provocation of muscle injury, were also performed within 30 of these other studies. Also, there were 4 studies with no specific stress test. It is interesting that for the Ubiquinol and Mitoquinone none of the exercise tests were similar.

Table 10 shows the data of the studies related to the sports level of the subject with the impact on physical and sports performance variables, inflammatory and oxidative patterns, muscle injury or other aspects of health, in a simple form, indicating whether it has a positive effect, no effect or a negative effect. As it is shown, there are more studies aimed at sporting people of a certain level, which in turn are those with the most positive results (39 vs. 12, 74.5%), with the data in relation to the oxidative and inflammatory response being the most attractive. No such benefit was observed in moderately active subjects, nor in non-active subjects (11 vs. 4; 73.3%). In a general observation of all subjects, there is a beneficial trend for all the evaluated objectives, sports performance, physical performance, inflammatory response pattern, oxidative and other health-related domains. On the other side, when age categories, Table 11, are evaluated, there is a trend of beneficial effect for all groups, with the most prominent positive effect on the oxidative pattern and the different health evaluations. The most evaluated group is <30 years old (63.8%), and the lowest is >50 years old (12.1%).

## 4. Discussion

Recent publications on CoQ_10_ supplementation suggest that it may be an interesting molecule in health [27,28] and for optimising exercise performance [32]. However, in the field of physical activity, there is a great diversity of work with different orientations, which sometimes makes it difficult to situate this work in the context of sports performance. Our discussion could provide a practical criterion to be taken into account by professionals wishing to assess this molecule (CoQ_10_) in the context of sport.

### 4.1. Dosage and Biodisponibility

Coenzyme Q_10_ is a water-insoluble substance with limited lipid solubility and a relatively high molecular weight, so it is also poorly absorbed in the gastrointestinal tract [16], but its reduced form (Ubiquinol) is 6–10 times more bioavailable than Ubiquinone or oxidised CoQ_10_ [100]. Its absorption and uptake pathways begin with emulsification and micelle formation with the fatty components of food, which is facilitated by pancreatic and biliary secretions [101]. Likewise, the efficiency of absorption also depends on the dose administered [102], the bioavailability [103,104] the diet [105] and the stability of the final product [106]. That is the reason why strict vegetarian diets may be deficient to maintain the CoQ_10_ level [107]. Not only because of the lower contribution of CoQ_10_, but also because the diet modifies its bioavailability. In the studies screened in this review, the dose of Ubiquinone administered is quite heterogeneous, ranging from 30 to 300 mg/day, with lower doses being more frequent in the older studies. With regard to Ubiquinol, the studies indicate doses ranging from 200 to 300 mg/day, with a single exception of 600 mg/day. In general, the studies analysed show a fivefold increase in plasma levels when Ubiquinol is administered at a dose of 250 mg/d and a twofold increase in plasma levels when Ubiquinone is administered at a dose of 150 mg/d.

The CoQ_10_ content in the body varies in different organelles, tissues and species [108]: in plasma, 0.46–1.78 μmol/L or 101–265 μmol CoQ_10_/mol Chol [109], and in muscle 140–580 nmol/g protein [110]. In football players, CoQ_10_ levels have been linked to sports performance [99]. In this regard, Bonetti et al. [40] indicate that structured exercise and training maintain and improve plasma levels over the long term. However, it should be noted that excessive physical work, illness and oxidative stress can lead to a reduction in plasma CoQ_10_ levels [51,73,98,111].

Throughout the analysis of the literature data used in this review, we found that the mean plasma CoQ_10_ level in high-training subjects was 0.97 μg/mL, compared to 0.83 μg/mL in moderately trained subjects or 0.93 μg/mL in sedentary subjects. Only two articles found higher levels, Tauler et al. [69]: 3 μg/mL, and Kizaki et al. [85]: 1.7 μg/mL. Laaksonen et al. [58] reported values of 0.9 μg/mL in young athletes and 1.3 μg/mL in older athletes. However, at the muscle level, the young athletes had a higher concentration of CoQ_10_, 118 μg/mL compared to 92 μg/mL in the older subjects. This discrepancy in the population of a certain age has already been assessed in plasma, irrespective of differences in formulations and excipients, doses administered and treatment time [112].

However, it also warns us that an increase in plasma does not necessarily correlate with an increase in muscle tissue. It is possible that sufficient treatment time or higher dosage may be required, as it occurs in animal models where it is shown that chronic ingestion of relatively large doses of CoQ_10_ in the diet is able to increase the CoQ_10_ concentrations, especially in the mitochondrial fractions of the heart [113]. It is therefore worth remembering that the use of the same dose of Ubiquinone, or Ubiquinol, taken orally by athletes with different body compositions, leads to different intakes of CoQ_10_ per kg of body weight [32]. On the other side, the change in muscle and plasma levels in relation to the CoQ_10_ administration is inconsistent. In this way some authors [45,58,68,78] have shown an increase of Q_10_ in plasma that it was not reflected in the muscle. In contrast, Laaksonen et al. [58] in the young athletes and Drobnic et al. [47] do observe an increase in concentration, although not significant, before and after supplementation. This discrepancy may be due to other methodological aspect, and to the CoQ_10_ regimen administered. Muscle biopsy is usually performed by needle biopsy. Although fine-needle aspiration is a well-referenced model, the samples are relatively small (15–30 mg Cooke, 40–80 mg Svensson, 50–100 mg Zhou), which are, in our case intended to illustrate the entire composition of the muscle mass of a high level or veteran athlete. This type of sample is undoubtedly excellent for discriminating and diagnosing pathology but limits the repeatability of assessments when quantifying molecules, so a larger muscle sample is much appreciated by the laboratory pathologist. In the last study discussed [47,98], a muscle biopsy sample of 200–250 mg was obtained from the vastus lateralis muscle using the ENCOR ULTRA^®^ 10 G biopsy system under ultrasound guidance. This is a simple and practical method, although not cheap, but it provides a high-quality homogeneous sample, very suitable for the study of muscle tissue, and allows portions to be retained for various subsequent analyses and pooling if a multicentre project is considered.

### 4.2. Antioxidant Activity

The use of CoQ_10_ for its antioxidant effect has been widely described in different pathologies [30,114,115,116,117] and in exercise [34,46,65,115,118]. The most important and relevant of its actions is the potent antioxidant capacity of its coexisting redox forms (ubiquinone, semiubiquinone and ubiquinol), which act on the mitochondrial and cell membrane [115,116,117,118]. These antioxidant properties of CoQ_10_ also help to recycle other antioxidants such as vitamin C and vitamin E, which act on free radicals or oxidants, reducing and neutralising compounds [116,117,118].

It is estimated that the increase in the production of free radicals and other ROS by exercise [119] accounts for 1% to 5% of the oxygen consumed during respiration [120]. ROS have been suggested to be the main reason for exercise-induced alterations in the redox balance of muscle, promoting oxidative injury and muscle fatigue [121], in addition to the damage to DNA and muscle structure [122], all of which impairs athletic performance. In addition, CoQ_10_ also regulates eNOS function in different cell membranes. Therefore, the depletion of CoQ_10_ may promote uncoupling of eNOS, making it an additional source of ROS, shifting the nitro-redox balance towards oxidation [123] that can be helpful at different ages [124].

As can be seen in Table 10, which shows the effects of Ubiquinone and Ubiquinol supplementation on different variables as a function of physical condition, antioxidant capacity in relation to exercise is rated as much more beneficial than the others. It is evident in both athletes and sedentary subjects irrespective of their age. This suggests that their antioxidant activity is real and may be aimed to prevent the excess of ROS favour the tissue and bioenergetic recovery of the cells.

### 4.3. Muscular Injury and Inflammatory Process

High intensity, strenuous and/or long-duration exercise, especially eccentric contraction exercise, can induce muscle damage (EIMD). This damage is derived from the mechanical stress caused and the inflammatory response [125,126,127,128,129]. This inflammatory response also leads to the release of ROS and cytokines [130,131] promoting the activation of gene transcription factors [132,133]. During the inflammatory process, there is increased leakage of muscle proteins into the circulation (creatine kinase-CK, myoglobin-Mb, aspartate transaminase-AST) [134,135,136,137,138]. The CK and Mb are indirect markers of muscle damage, moreover, ROS produced by neutrophils contribute to muscle damage and circulating neutrophils increase after exercise, leading to muscle damage [127,138].

The effect of CoQ_10_ supplementation on exercise-induced muscle damage CK level is controversial. It decreases in rats [139], but the results in the lab for humans are very diverse, offering similar [64,65,79], lower [42,48,49,86,90] or even higher [60] concentrations. In the case of the field tests in runners, nor Kaikkonen et al. [58] nor Armanfar et al. [39]) did observe any change on CK activity after a marathon or a 3000 m test, respectively.

Of all these biomarkers, CK is the most commonly used to assess muscle damage [140,141,142,143,144]. The CK enters the blood through the increase in membrane permeability that occurs after exercise. In addition, plasma CK amounts are conditioned by many exercise-related factors, such as intensity, duration, training status of the individual or exercise experience [141,142]. The amount of muscle mass and CK content within the muscle fibre, which is dependent on individual genetics and gender, must also be considered [143,144,145]. Moreover, muscle damage shows an enormous inter-individual variability that depends on additional factors that could exert an additional influence [143,146,147,148,149,150,151] so its use should not be used as a gold standard parameter for assessing the extent of muscle injury due to exercise, as its use is controversial for research [152] or monitoring training [153] but it can be useful as a complementary value associated with other more specific determinations.

The impact of CoQ_10_ in the prevention of muscle injury should lie mainly in its antioxidant and anti-inflammatory action, its protective action on mitochondria and DNA and its modifying effect on gene expression [154]. In addition, CoQ_10_ stabilises the phospholipid structure of cell membranes and protects skeletal muscle cells [155]. The CoQ_10_ supplementation may therefore reduce exercise-induced muscle injury by increasing the concentration of CoQ_10_ in muscle cell membranes and stabilising the cell membrane.

Other authors [156] have reported that CoQ_10_ treatment attenuates the oxidative activity of neutrophils. Thus, as can be seen in this review, the studies analysed show a beneficial tendency for the administration of CoQ_10_ to modify the inflammatory response, cytokines and transfer factors [39,46,47,84,89].

### 4.4. Sport and Exercise Performance

The bioenergetic approach to sport performance is one of many different ways to assess the success of exercise performance [157]. The determination of the maximal oxygen uptake (VO_2_max), anaerobic threshold, lactate concentration, presence of CK, oxidation-reduction metabolism molecules, perception of effort, heart rate evolution or fatigue index are some of the other variables frequently studied in research on the supplementation of ergogenic aids.

The evaluation of the VO_2_max is undoubtedly one of the fundamental variables when it comes to observing the effect of a exercise-enhancing substance. However, oxygen consumption can be limited by several factors with all the systems involved: respiratory, haematological, circulatory and mitochondrial metabolism [158]. On the other side, it is the energy cost that really “makes the difference” [159]. Coenzyme Q_10_, has some place in that environment, and therefore must be considered, as it is certainly in ATP regeneration and anaerobic threshold conflict that the useful administration of a compound is likely to have an influence. The VO_2_max will determine exercise time to exhaustion, so we may see great variability, which should be taken into account when considering a study on the effect of ergogenic aids. In this regard, McLellan et al. [160] have calculated a range of variability from 2.8 to 31.4% for five submaximal tests in cyclists during exercise to exhaustion. Also, Pereira et al. [161] have observed that the intraindividual variation of running economy cannot be underestimated, mostly when a substance used to enhance the success is under evaluation.

The ability to maintain a high percentage of maximal oxygen uptake offers the possibility of achieving high exercise intensity [162]. Although the gas exchange threshold (GET) is a good predictor of exercise performance, it does not reflect the specific intensity of the athlete’s competition [163]. Alternatively, other variables such as critical power (CP) (for cycling) or critical velocity (for running) have emerged as more viable surrogates and are considered to be more consistent in high-intensity exercise [164]. Indeed, CP represents the highest oxygen volume at which lactate and blood oxygen levels can stabilize [165]. Because of this, CP has been found to be a robust parameter representative of a fatigue threshold, located approximately midway between GET and VO_2_max., which demarcates strong from severe intensity domains [162].

Only in two studies, out of the nine selected in this review, is there some evidence of improved physical performance attributable to CoQ_10_ supplementation. With respect to the 55 initially screened, most of them assess variables related to maximal work or exercise levels reaching fatigue. However, it is not easy to justify a possible VO_2_max improvement effect when this depends on many factors as it has exposed [158] other than those provided by CoQ_10_ supplementation, especially in subjects whose mitochondria are not primed to be significantly more efficient without an adequate period of associated stimulus (exercise). Mitochondrial capacity appears to be much higher than that assessed by this parameter [165]. Individuals with higher mitochondrial density enjoy a higher margin of functional mitochondrial capacity, which perhaps explains why is greater the effect of COQ_10_ when administered to them. It should be remembered that mitochondrial capacity may be underexpressed in individuals who, although not physically active at the start of the studies, were very active in their youth and have a pool of latent mitochondria that have not disappeared [166]. That would be also the case for athletes who have taken anabolic steroids when young [167]. The increased mitochondrial responsiveness can be activated with an appropriate physical and nutritional stimulus. It is quite possible that the change in performance would be different in individuals who never played sport intensively in their youth than those who did. In any case, the results would be homogenised across studies if mitochondrial density were determined.

### 4.5. Other Action of Coenzyme Q_10_ Associated to Exercise

Of the selected studies, two explore the exercise-associated effect of CoQ_10_ on glucose metabolism and bone remodelling [56,84]. Interestingly, as CoQ_10_ is a ubiquitous molecule at the metabolic level, it is expected to have positive effects by optimising the response of exercise-related systems [42,71,79,87].

There are many other actions or possible implications of this molecule in relation to exercise and sport [32]: its impact on the nervous system and muscle diseases [168], stabilizing red blood cells (to increase the resistance to oxidative stress) [169], more fluidity properties [170], optimizing endothelial dysfunction [171,172], and even modifying muscle composition [173]. All of which are extremely interesting and which, once studied in-depth, could determine the true impact of CoQ_10_ on each of them.

Furthermore, CoQ_10_ supplementation may work synergistically with other molecules that help to prevent or restore tissue after the stress produced by the exercise. This is the case of creatine in the energy metabolism and tissue restoration [174,175,176], the omega-3 fatty acids in their antioxidant and cellular modulating activity [177,178,179], or the curcumin for its specific antiinflammatory and antioxidant activity [180,181,182].

## 5. Recommendations for Future Research

One of the major limitations is that there is a diversity of exercise modalities in different sports disciplines. Additionally, the sample size of the studies is not homogeneous. Moreover, some of the studies do not specify gender and there is also the problem of doses and times of use of CoQ_10_ supplementation.

Therefore, we believe that given the differences between the studies evaluated, we can offer a reference of methodological guidelines with a certain justification to facilitate a more precise analysis of those that have been developed with scientific guarantees in the future (Table 12).

The rationale for each of the recommendations is the subject of specific comments in the discussion of this review, such as the desire that it should not be a crossover study or that the exercise variables to be determined before and after supplementation should be well-defined. Others are basic, such as using placebos, or suggesting that the study be double-blind or randomised [183,184]. However, there are others that are set out in the table and which we extend below.

There are four types of populations that are markedly different in the possible action of CoQ_10_ in relation to physical performance: the sedentary, the moderately physically active and the very active [185,186], which in turn correspond to different age groups. Functionality, learning, memory and mitochondrial density are likely to behave differently after exposure to the supplement. The proposed table shows four age ranges, which are somewhat arbitrary and could be adjusted to human age periods [187], but since these correspond more to concepts of mental maturity and senility, it is considered more interesting to associate them with aspects of physical maturation and exposure to pathology [188] rather than purely psychological or mental aspects.

The administered dose of both the oxidised and reduced forms must be adequate to ensure that bioavailability is correct. A dose is proposed that, based on the latest studies with Ubiquinol, shows an adequate increase in plasma levels. Moreover, the period of administration must be given a minimum of time to justify that the functional results correspond to the presence of the element studied in the place where it functions, the mitochondria [101,108]. This thinking justifies the determination of levels not only in plasma, which we consider mandatory, but also recommends determination in muscle tissue to associate it with the mitochondrial characteristics of interest to us. In addition, it is recommended that at least one biopsy be taken from the vastus lateralis quadriceps if it is decided to do several. The different concentration in humans is not fully defined but it is presumable that it occurs as in rodents and its concentration is different depending on the muscle examined [188,189]. In turn, plasma determination should not be done when the treatment has been administered at a time when plasma availability is high due to absorption. It is therefore justified that at least 24 h should elapse, considering the bioavailability of the substance [16].

It is also necessary to consider aspects related to the idiosyncrasies of the subject in relation to the group studied, such as the diet maintained, excessive exercise, or the sedentary lifestyle of adults who practised high-intensity sport in their youth, as explained in the discussion.

## 6. Conclusions

From the evaluation of the various studies reviewed, and considering the methodological difficulties that exist between them, it can be concluded that the use of Coenzyme Q_10_ seems to offer a good profile in the control of an oxidative pattern with a certain anti-inflammatory activity at the cellular level in response to exercise in the various populations studied, in addition to some properties that should be studied in greater depth. It can therefore be seen as a protective and recuperative substance rather than an ergogenic substance in itself. Coenzyme Q_10_ is a promising molecule in terms of its qualities and safety profile, which is why it is considered necessary to carry out studies with a methodology that is more oriented towards the profile of the substance under study, with practical criteria that could favour the performance of more complex analyses between the various scientific groups involved.

## Figures and Tables

**Figure 1 nutrients-14-01811-f001:**
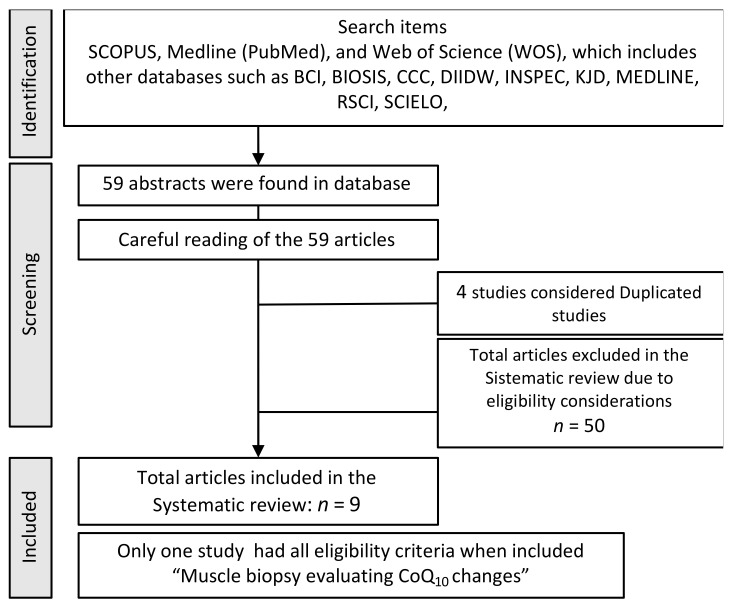
Full search strategy. Eligibility criteria of the review: randomised, double-blind, placebo-controlled, parallel design, determining at least plasma CoQ_10_ levels, and that the human sample were athletes or subjects trained and used to practising high-intensity exercise.

**Table 1 nutrients-14-01811-t001:** Methodological quality of the studies included in the systematic review. Assessment of the methodological quality of studies using the PEDro Scale.

References	Svensson et al. [68]	Díaz-Castro et al. [83,84]	Kizaki et al. [85]	Kon et al. [86]	Sarmiento et al. [89]	Suzuki et al. [90]	Braun et al. [41]	Ho et al. [56]	Weston et al. [72]
Criteria 1 *	1	1	1	1	1	1	1	1	1
Criteria 2	0	1	1	1	1	1	1	1	1
Criteria 3	1	1	1	1	1	1	1	1	1
Criteria 4	1	1	0	0	1	1	1	1	0
Criteria 5	1	1	1	1	1	1	1	1	1
Criteria 6	1	1	1	1	1	1	1	1	1
Criteria 7	1	1	1	1	1	1	1	1	1
Criteria 8	1	1	1	1	0	0	1	1	1
Criteria 9.	1	1	1	1	1	1	1	1	1
Criteria 10	1	1	1	1	1	1	1	1	1
Criteria 11	1	1	1	1	1	1	1	1	1
PEDro Score	9	10	9	9	9	9	10	10	9

Criteria 1 * = The eligibility criterion is not scored for being related to external validity and therefore does not reflect the quality dimensions assessed by the PEDro Scale.

**Table 2 nutrients-14-01811-t002:** Studies with the inclusion criteria.

Reference		Molecule	CoQ_10_ mg/d	Duration (d: Days)	Placebo	n Total/CoQ_10_	Type of Subjects	Sex	Sport/Activity	Exercise Testing	Age (Years)	Impact on Phys./Sport Perf.
Svensson et al. (1999)	[68]	Ubiquinone	120	20 d	Yes	17/9	Well trained subjects	Male	-	Graded max and Anaerobic tests	28 ± 5	No
Díaz-Castro et al. (2020)	[83,84]	Ubiquinol	200	14 d	Yes	100/50	Well trained subjects	Male	Firemen	Circuit edurance exercises	39 ± 1	-
Kizaki et al. (2015)	[85]	Ubiquinol	600	11 d	Yes	32/17	Well trained subjects	Male	Kendo	4 training days	20 ± 1	No
Kon et al. (2008)	[86]	Ubiquinol	300	20 d	Yes	18/10	High level	Male	Kendo	Muscle injury induced exercise	20 ± 1	Yes
Sarmiento et al. (2016)	[89]	Ubiquinol	200	14 d	Yes	100/50	Well trained subjects	Male	Firemen	Circuit edurance exercises	39 ± 9	-
Suzuki et al. (2021)	[90]	Ubiquinol	300	12 d	Yes	16/8	Well trained subjects	Male	Distance runners	25 & 40 K races	20 ± 2	Yes
Braun et al. (1991)	[41]	Ubiquinone	100	56 d	Yes	12/6	High level	Male	Cyclists	Graded max test	22 ± 2	No
Ho et al. (2020)	[56]	Ubiquinone	300	84 d	Yes	31/15	Moderately trained subjects	Male & Female	Soccer & Taekwondo	None	20 ± 1	-
Weston et al. (1997)	[72]	Ubiquinone	≃70	28 d	Yes	18/6	High level	Male	Cyclists and Triathletes	Graded max.	25 ± 3	No

**Table 3 nutrients-14-01811-t003:** Studies with the inclusion criteria.

Reference	Change CoQ_10_ Total				Measured Parameters and Effects of CoQ_10_					
	Tissue	Plasma Pre ➝ Post (μg/mL)	Plasma Pre ➝ Post (μmol/mol Chol)	Muscle Pre ➝ Post (nmol/g Protein)	Inflammatory Pattern	Antioxidant Pattern	Physical Performance	Sport Performance	Muscle Injury	Other
Svensson et al. (1999)	Plasma ↑ Muscle ↔	0.74 ➝ 1.23		40.8 ➝ 44.2 mgl/kg		HX, MDA, UA ↔				
Díaz-Castro et al. (2020)	Plasma ↑	1.00 ➝ 5.22			VEGF, NO, EGF, IL-1^ra^, IL-10 ↑, and IL-1, IL-8, MCP-1 ↓					PTH, OC, OPG, phosphatase al., leptin, insulin, noradrenaline and PGC-1α ↑,
Kizaki et al. (2015)	Plasma↑	0.7 ➝ 10.8							CK, Mb ↔,	
Kon et al. (2008)	Plasma↑	≃0.8 ➝ ≃3.8				LPO ↓			CK, Mb ↓	
Sarmiento et al. (2016)	Plasma ↑	0.9 ➝ 4.5			NO ↓	hydroperoxydes Isoprostanes, oxidized LDL, TAC ↑				
Suzuki et al. (2021)	Plasma ↑	0.7 ➝ 5.6					Perception of fatigue ↓		CK, ALT, LDH ↓	
Braun et al. (1991)	Plasma ↑	≃0.8 ➝ 1.5				MDA ↓ (NS)	Total work load, VO_2_peak, HR ↔			CoQ10 postexercise ↑
Ho et al. (2020)	Plasma ↑	0.57 ➝ 1.14	130 ➝ 270			MDA ↓, TAC ↔				Improve glycemic control
Weston et al. (1997)	Plasma ↑	0.9 ➝ 2.0					Oxygen uptake at 6 min ↑, VO_2_max, max power, Anaerobic threshold, and HRTime ↔			

CoQ_10_/Chol: CoQ_10_ in relation to cholesterol level; pre/post: previous and post supplementation; Ns: No specified; Phys. Physical; Perf: Performance; ↔ similar response to placebo or previous supplementation, ↑ Higher response to the presupplementation phase, ↓ Lower response to the presupplementation phase. HX hypoxanthine, MDA malondyaldehide, UA uric acid, PTH parathormone, OC osteocalcin, OPG osteoprotegerin, PGC-1α peroxisome proliferator activated receptor-γ coactivator-1α VEGF vascular endothelial growth factor, NO nitric oxide, EGF epidermal growth factor, CK creatinkinase, Mb myoglobin, ALT alanine transaminase, LDH lactate dehydrogenase, TAC Total antioxidant capacity, MCP-1 monocyte chemotactic protein, LPO lipid peroxidation.

**Table 4 nutrients-14-01811-t004:** Studies without the inclusion criteria.

Reference		Molecule	CoQ_10_ mg/d	Duration (d: Days)	Placebo	n Total/CoQ_10_	Type of Subjects	Sex	Sport/Activity	Exercise Testing	Age (Years)	Impact on Phys./Sport Perf.
Alf et al. (2013)	[81]	Ubiquinol	300	42 d	Yes	100/50	High level	Male & Female	Olympic Athletes	Cycling submaximal test	19 ± 3	Yes/Yes
Bloomer et al. (2012)	[82]	Ubiquinol	300	28 d	Yes	15/15	Well trained subjects	Male & Female	Ns	Graded and Anaerobic max. Tests	43 ± 10	No
Kunching et al. (in press)	[87]	Ubiquinol	200	42 d	Yes	29/15	Moderately trained subjects	Male	Diverse	Indirect max. Test, 1RM and flexibility	24 ± 2	Yes
Orlando et al. (2018)	[88]	Ubiquinol	200	28 d	Yes	21/21	Moderately trained subjects	Male	Rugby	40 min 85%max. Treadmill	26 ± 5	No
Pham et al. (2020)	[80]	Ubiquinol	200	42 d	No	22/22	Physically active subjects	Male	-	None	51 ± 1	-
Broome et al. (2021)	[93]	Mitoquinone	20	28 d	Yes	19/19	Well trained subjects	Male	Cyclists	8 km race	44 ± 4	Yes
Pham et al. (2020)	[80]	Mitoquinone	20	42 d	No	22/22	Physically active subjects	Male	-	None	51 ± 1	-
Shill et al. (2021)	[91]	Mitoquinone	10	21 d	Yes	20/10	Physically active subjects	Male	-	Graded max test	22 ± 1	No
Williamsom et al. (2020)	[92]	Mitoquinone	20	21 d	Yes	24/12	Physically active subjects	Male	-	Anaerobic repeated tests	25 ± 4	Yes
Amadio et al. (1991)	[38]	Ubiquinone	100	40 d	Control	10/5	Well trained subjects	Male	Basketball	Submaximal test	19 ± 5	Yes
Armanfar et al. (2015)	[39]	Ubiquinone	300–400	14 d	Yes	18/9	Well trained subjects	Male	Middle distance runners	3000 m race	20 ± 3	No
Bonetti et al. (2000)	[40]	Ubiquinone	100	56 d	Yes	28/14	Moderately trained subjects and NA	Male	Cyclists	Graded max test	41 ± 6	Yes
Cerioli (1991)	[42]	Ubiquinone	100	30 d	Ns	12	Non active	Male	-	Graded max test	26	Yes
Cinquegrana et al. (1987)	[43]	Ubiquinone	60	35 d	Yes	14/14	Non active	Male	-	Graded max test	48 ± 3	
Ciocoi-Pop et al. (Note I & II) (2009)	[44]	Ubiquinone	30	21 d	Yes	10/5	Well trained subjects	Male	Soccer	Graded max and Anaerobic tests	19 ± 0	Yes
Cooke et al. (2008)	[45]	Ubiquinone	200	14 d	Yes	31/21	Moderately trained subjects and untrained	Male & Female	-	Grade max, Isokinetic, and Anaerobic tests	26 ± 8	No
Díaz-Castro et al. (2012)	[46]	Ubiquinone	30 × 2 days, 120 day test	3 d	Yes	20/10	Well trained subjects	Male	-	50 km running	41 ± 3	Yes

**Table 5 nutrients-14-01811-t005:** Studies without the inclusion criteria.

Reference	Change CoQ_10_ Total				Measured Parameters and Effects of CoQ_10_					
	Tissue	Plasma Pre ➝ Post (μg/mL)	Plasma Pre ➝ Post (μmol/mol Chol)	Muscle Pre ➝ Post (nmol/g Protein)	Inflammatory Pattern	Antioxidant Pattern	Physical Performance	Sport Performance	Muscle Injury	Other
Alf et al. (2013)	-						Power output ↑			
Bloomer et al. (2012)	Plasma ↑	0.98 ➝ 2.33	0.48 ➝ 1.13			MDA, hydrogen peroxide,	Lactate ↔			perceived vigour ↔
Kunching et al. (in press)	-						VO_2_max ml/kg ↑,Max strength ↔			Sistolic pressure ↓
Orlando et al. (2018)	Plasma ↔ CoQ10/Chol ↑		≃180 ➝ >500			ROS ↓	Time to exhaustion, speed ↔		CK, DNA damage ↔	
Pham et al. (2020)	-					H2O2 mit ↓, Isoprost ↔				
Broome et al. (2021)	-					ROS, Isoprost ↓	Power output ↑	Faster time trial.		
Pham et al. (2020)	-					H2O2 mit ↓, TAC ↑, Isoprostanes ↔				
Shill et al. (2021)	-					CDseries, VEGFR2+ and peripheral blood mononuclear cells ↔	VO_2_max ↔			Muscle mitochondrial capacity ↔
Williamsom et al. (2020)	-								DNA damage ↓	
Amadio et al. (1991)	Plasma ↑	0.9 ➝ 1.6					VO_2_max ↑			Cardiac parameters
Armanfar et al. (2015)	-				TNFa, CRP, IL6 ↓				CK ↔	
Bonetti et al. (2000)	Plasma ↑	0.8 ➝ 2.2				hypoxanthine, xanthine and inosine ↔	Max work load ↑ VO_2_peak, anaerobic threshold and lactate ↔,			
Cerioli (1991)	-						Aerobic Capacity↑		CK ↔	FFA ↓,Fat metabolism ↑
Cinquegrana et al. (1987)	-									
Ciocoi-Pop et al. (Note I&II) (2009)	-					MDA ↑, HD ↑ in saliva	VO_2_max ↑, Anaerobic power ↔			
Cooke et al. (2008)	Plasma ↑ Muscle ↔	≃0.6 ➝ ≃2.5		≃1.2 ➝ ≃1.4 μg/mg		MDA ↑, SOD ↓	VO_2_max, anaerobic capacity, Anaerobic power ↔			
Díaz-Castro et al. (2012)	-				IL-6 ↔, 8-OH-dG, TNF-α ↓	CAT ↑, TAS ↑, GPx ↔, hydroperoxide ↓, isoprostane ↓,				

CoQ_10_/Chol: CoQ_10_ in relation to cholesterol level; pre/post: previous and post supplementation; Ns: No specified; Phys. Physical; Perf: Performance; ↔ similar response to placebo or previous supplementation, ↑ Higher response to the presupplementation phase, ↓ Lower response to the presupplementation phase. GPx Glutathione peroxidase, MDA malondyaldehide, ROS reactive oxygen species, CK creatinquinase, VO_2_ oxygen consumption, HD hydrogen donors, SOD superoxide dismutase, CAT catalase, TAS Plasma total antioxidant status, 8-OH-dG 8-Hydroxy-20-deoxyguanosine, TNF-α Tumor necrosis factor a, FFA free fatty acids, CRP C-reactive protein, IL6 interleukin 6.

**Table 6 nutrients-14-01811-t006:** Studies without the inclusion criteria.

Reference		Molecule	CoQ_10_ mg/d	Duration (d: Days)	Placebo	n Total/CoQ_10_	Type of Subjects	Sex	Sport/Activity	Exercise Testing	Age (Years)	Impact on Phys./Sport Perf.
Drobnic et al. (2020)	[47]	Ubiquinone	100	30 d	Control	20/12	Well trained subjects	Male	Marathon runners	Treadmill Hot & Humid environment	55 ± 4	Yes
Emami et al. (2018)	[48]	Ubiquinone	300	14 d	Yes	36/9	Well trained subjects	Male	Swimmwers	Swimming training	18 ± 1	
Fiorella et al. (1991)	[49]	Ubiquinone	100	40 d	Control	22/11	Well trained subjects	Male	Athletes	Graded max test + Incremental running test until exhaustion	29 ± 6	Yes
García-Verazaluce et al. (2015)	[50]	Ubiquinone +	120 + Phlebodium d.	28 d	Yes	30/10	Well trained subjects	Male	Volleyball	None	25 ± 2	Ns
Geiss et al. (2004)	[51]	Ubiquinone	180	28 d	Yes	10/10	Well trained subjects	Male?	Endurance	Submaximal fatigue test	Ns	Yes
Gökbel et al. (2010)	[52]	Ubiquinone	100	56 d	Yes	15/15	Non active	Male	Ns	Anaerobic test(Repeated Wingate test)	20 ± 1	No
Gökbel et al. (2016)	[53]	Ubiquinone	200	98 d	Yes	23/23	Patients (hemodyalisis)	Male	Ns	6 Mins walking test	47 ± 12	No
Guerra et al. (1987)	[54]	Ubiquinone	60	35 d	Ns		Moderately trained subjects	Male	Cycling	Graded max test. Race 9 km.	Ns	Yes/Yes
Gül et al. (2011)	[55]	Ubiquinone	100	56 d	Yes	15	Non active	Male	-	Anaerobic repeated tests	20 ± 1	Yes
Kaikkonen et al. (1998)	[57]	Ubiquinone +	90 + Vit E	21 d	Yes	37/18	Moderately trained subjects	Male	Marathon	Marathon	40 ± 7	No
Laaksonen et al. (1995)	[58]	Ubiquinone +	120 + Omega 3	42 d	Yes	11/11	Well trained subjects	Male Male	Marathon & triathletes	Graded max. test	28 (22–38)	No
						8/8					64 (60–74)	No
Leelarungrayub (2010)	[59]	Ubiquinone	300	12 d	No	16/16	Moderately trained subjects	Male & Female	Swimmimg	Treadmill time to exhaustion & Swimmimg 100–800 m	15 ± 1	Yes 100 m, No 800 m
Malm et al. (1996)	[60]	Ubiquinone	120	20 d	Yes	15/9	Moderately trained subjects	Male	-	Anaerobic tests	20–34	No
Malm et al. (1997)	[61]	Ubiquinone	120	22 d	Yes	18/9	Moderately trained subjects	Male	-	Graded max and Anaerobic tests	25 ± 3	No
Mizuno et al. (2008)	[62]	Ubiquinone	100 & 300	56 d	Yes	17/17	Physically active subjects	Male & Female	-	Anaerobic repeated tests under fatigue	38 ± 10	Yes
Nielsen et al. (1999)	[63]	Ubiquinone +	100 + Vit E & Vit C	42 d	Yes	7/7	Well trained subjects	Male	Triathletes	Graded max. & Local fatigue (31P-NMRS)	22–32	No

+: indicates that the administration of Ubiquinone is supplemented by one or more other substances (indicated in the next column).

**Table 7 nutrients-14-01811-t007:** Studies without the inclusion criteria.

Reference	Change CoQ_10_ Total				Measured Parameters and Effects of CoQ_10_					
	Tissue	Plasma Pre ➝ Post (μg/mL)	Plasma Pre ➝ Post (μmol/mol Chol)	Muscle Pre ➝ Post (nmol/g Protein)	Inflammatory Pattern	Antioxidant Pattern	Physical Performance	Sport Performance	Muscle Injury	Other
Drobnic et al. (2020)	Plasma ↑ Muscle ↑	1.11 ➝ 2.34	212 ➝ 476	245 ➝ 299nmol/g protein	IL-6, IL-8, IL-10, MCP-1, TNFa ↓	MDA ↓, TAC ↑ after exercise	Lactate and fatigue perception ↑,			
Emami et al. (2018)	Plasma ↑	≃0.8 ➝ ≃2.5				TAC ↑, LPO ↓,			LDH, CK-MB, Mb, Troponin I ↓	
Fiorella et al. (1991)	Plasma ↑ Thrombocites↑	0.6 ➝ 1.4 37.5➝61.8					LA, UA, Ammonia ↔	Running distance and time to exhaustion ↑,	CK, LDH ↓,	
García-Verazaluce et al. (2015)	-				IL6 ↓					Costisol ↓
Geiss et al. (2004)	Plasma ↑	0.6 ➝ 1.7					Power output ↑			
Gökbel et al. (2010)	-						Peak Power ↔, Mean Power ↑, Fatigue Index ↔			
Gökbel et al. (2016)	Plasma ↑	1.3 ➝ 3.0				MDA ↓, GPX↓, SOD ↔, after exercise (NS)				
Guerra et al. (1987)	Plasma ↑	Ns					VO_2_max ↑	Possible better race time		
Gül et al. (2011)	-					MDA ↓, NO, XO, SOD, GPx ↔, UA ↑				
Kaikkonen et al. (1998)	Plasma ↑	1.96 ➝ 2.03				GSH, UA, LDLox, TRAP ↔				
Laaksonen et al. (1995)	Plasma ↑ Muscle ↑	0.9 ➝ 2.0		118 ➝ 128 nmol/g protein		MDA ↔	VO_2_max ↔, Time to exhaustion			
	Plasma ↑ Muscle ↔	1.31 ➝ 3.5		92 ➝ 78 nmol/g protein						
Leelarungrayub et al. (2010)	Plasma ↑	1.1 ➝ 2.3				MDA, NO ↓, TAC, SOD ↔, GSH ↑	Increase time to fatigue	No better 800 m swimming time		
Malm et al. (1996)	-								CK ↑	
Malm et al. (1997)	-					ROS ↑	VO_2_max ↔, Max power ↔			
Mizuno et al. (2008)	Plasma ↑	100 mg. 0.5 ➝ 2.0 300 mg. 0.5 ➝ 3.3					Perception of fatigue ↓, Perception of recovery, Max velocity ↑			
Nielsen et al. (1999)	Plasma ↑	0.9 ➝ 1.8					VO2max ↔			Muscle metabolism ↔

CoQ10/Chol: CoQ10 in relation to cholesterol level; pre/post: previous and post supplementation; ➝: Concentration change; Ns: No specified, Phys. Physical; Perf: Performance; ↔ similar response to placebo or previous supplementation, ↑ Higher response to the presupplementation phase, ↓ Lower response to the presupplementation phase, NS: Non statistical significance. MDA malondyaldehide, TAC total antioxidant capacity, LPO lipid peroxidation, LDH lactate dehydrogenase, CK-MB miocardic creatinquinase, Mb myoglobin, LA lactic acid, UA uric acid, CK creatinquinase, SOD superoxide dismutase, NO nitric oxide.

**Table 8 nutrients-14-01811-t008:** Studies without the inclusion criteria.

Reference		Molecule	CoQ_10_ mg/d	Duration (d: Days)	Placebo	n Total/CoQ_10_	Type of Subjects	Sex	Sport/Activity	Exercise Testing	Age (Years)	Impact on Phys./Sport Perf
Okudan et al. (2017)	[64]	Ubiquinone	200	28 d	Yes	21/11	Non active	Male	-	Exccentric ex.	23 ± 0	No
Östman et al. (2012)	[65]	Ubiquinone	90	56 d	Yes	23/11	Moderately trained subjects	Male	-	Various exercise capacity tests	28 ± 9	No
Porter et al. (1995)	[66]	Ubiquinone	150	56 d	Yes	13/6	Physically active subjects	Male	Some with hypertension	Graded Max and Forearm Handgrip tests	45 ± 2	Yes
Snider et al. (1992)	[67]	Ubiquinone +	100 + Vit E, C inosine, citochrome C	28 d	Yes	11/11	High level	Male	Triathletes	Graded max. test	25 ± 1	No
Tauler et al. (2008) Ferrer et al. (2009)	[69,70]	Ubiquinone +	100+ Multivitamin	90 d	Yes	19/8	High level	Male	Soccer	Competition Match	20 ± 0	Yes
Vanfraechem et al. (1981)	[71]	Ubiquinone	60	56 d	Yes	6	Non active	Male	None	Graded max. 4w–8w	22 ± 2	Yes
Wyss et al. (1990)	[73]	Ubiquinone	100	30 d	Yes	18/18	Physically active subjects	Male	Running	Graded max.	25 ± 4	Yes
Yamabe et al. (1991)	[74]	Ubiquinone	90	6 months	No	9/9	Non active with inabilities to do exercise	Male	-	Graded max.	51 ± 5	Yes
Ylikoski et al. (1997)	[75]	Ubiquinone	90	42 d	Yes	18/18	High level	Male	cross-country skiers	Graded max.	Ns	Yes
Zeppilli et al. (1991)	[76]	Ubiquinone	100	30 d	Yes	9/9	High level	Male	Volleyball	Graded max test	17–2	Yes
						10	Non active		-		23–29	Yes
						8	Patients		Mit. Disease		23–29	Yes
Zheng et al. (2008)	[77]	Ubiquinone	30	1d	Yes	11/11	Non active	Male	-	Rest & Low intensity exercise (30% HRmax)	26 ± 1	Yes
Zhou et al. (2005)	[78]	Ubiquinone +	150+ Vit E	14 d	Yes	6/6	Physically active subjects	Male	-	Submaximal exercise and Graded max tests	30 ± 7	No
Zuliani et al. (1989)	[79]	Ubiquinone	100	28 d	No	12	Non active	Male	-	60’ ciloergometry	26	No
Sanchez-Cuesta et al. (2020)	[99]	.	-	2 Sport seasons	-	24 & 25	High level	Male	Soccer	Weekly competition	26 ± 4	Yes

+: indicates that the administration of Ubiquinone is supplemented by one or more other substances (indicated in the next column).

**Table 9 nutrients-14-01811-t009:** Studies without the inclusion criteria.

Reference	Change CoQ_10_ Total				Measured Parameters and Effects of CoQ_10_					
	Tissue	Plasma Pre ➝ Post (μg/mL)	Plasma Pre ➝ Post (μmol/mol Chol)	Muscle Pre ➝ Post (nmol/g Protein)	Inflammatory Pattern	Antioxidant Pattern	Physical Performance	Sport Performance	Muscle Injury	Other
Okudan et al. (2017)	Plasma ↑	1.0 ➝ 1.7				MDA, SOD ↔			CK, Myogb. ↔	
Östman et al. (2012)	-					MDA, UA, hypoxanthine ↔	Exercise capacity, VO_2_max, Max power, Lactate ↔		CK, UA ↔	
Porter et al. (1995)	Plasma ↑	0.7 ➝ 1.0					VO_2_max ↔ LA ↓(NS)			Vigor perception ↑
Snider et al. (1992)	-						Time to exhaustion, RPE ↔			
Tauler et al. (2008) Ferrer et al. (2009)	Plasma ↑	3.0 ➝ 3.6				SOD ↑, MDA ↓		↑ Time spend in active working zone (Z3-Z5)		
Vanfraechem et al. (1981)	-						Maximal load and O_2_ consumption 170 w and max.			Cardiaovascular and cardiorespiratory parameters ↑
Wyss et al. (1990)	Plasma ↑	Ns					VO_2_max ↑, Max work ↑,			
Yamabe et al. (1991)	-						VO_2_max, Max power and Anaerobic threshold ↑			
Ylikoski et al. (1997)	Plasma ↑	0.8 ➝ 2.8					VO_2_max, max power, Anaerobic threshold, and HRTime↑			
Zeppilli et al. (1991)	Plasma ↑	0.6 ➝ 1.3 Vb 0.7 ➝ 1.0 NA					VO_2_max, max power ↑			
Zheng et al. (2008)	-						VCO_2_/VO_2_ ↓, HR ↔			↑ Power HR variability
Zhou et al. (2005)	Plasma↑, Muscle ↔	0.8 ➝ 2.6		207 ➝ 220 (nmol/g protein)			VO_2_max ↔, HR ↔, RPE ↔, Anaerobic threshold ↔			
Zuliani et al. (1989)	Plasma ↑	0.5 ➝ 1.3					Glucose, Insulin, LA ↔		CK ↔,	Glycerol ↔, FFA ↓
Sanchez-Cuesta et al. (2020)	Plasma ↑	Preseason 0.6 Middle season 0.9					The highest values have better competition parameters		CK ↓	Better Testosterone/cortisol pattern

CoQ_10_/Chol: CoQ_10_ in relation to cholesterol level; pre/post: previous and post supplementation; Ns: No specified; Phys. Physical; Perf: Performance; ↔ similar response to placebo or previous supplementation, ↑ Higher response to the presupplementation phase, ↓ Lower response to the presupplementation phase, NS: Non statistical significance. The study of Sanchez-Cuesta as it is explained in the text, is the follow up of the CQ_10_ plasma level throughout 2 competition seasons. Vb volleyball, NA No physical activity, HR heart rate, RPE rating perceived perception.

**Table 10 nutrients-14-01811-t010:** Results of the Ubiquinone and Ubiquinol supplementation on different sport and exercise variables depending on the physical condition of the subjects evaluated.

Physical Condition Categories	Number of Studies	CoQ_10_ Effect	Total	Sport Performance	Exercise Performance	Oxidative Pattern	Muscle INJURY	Inflammatory Pattern	Other
High level/Well trained athletes	28	Positive	39	3	12	10	4	6	4
		No effect	12		6	1	3	1	1
Moderate trained/Physically active	20	Positive	11		4	5	1		2
		No effect	18	1	9	5	3		
		Negative	3			2	1		
Non active subjects	9	Positive	11		6	1			4
		No effect	4		2		2		
Patients	1	Positive	2		1	1			
TOTAL	58	number tests	100 *	3	40	25	14	7	11
		Positive	63	2	23	17	5	6	10
		No effect	34	1	17	6	8	1	1
		Negative	3			2	1		

* In the Zeppilli research [77] there are three populations (<30 a) that are studied with the same methodology. That is the reason why there are 2 more tests than in Table 11.

**Table 11 nutrients-14-01811-t011:** Results of the Ubiquinone and Ubiquinol supplementation on different sport and exercise variables depending on the age categories of the subjects evaluated.

Age Categories	Number of Studies	CoQ_10_ Effect	Total	Sport Performance	Exercise Performance	Oxidative Pattern	Muscle Injury	Inflammatory Pattern	Other
<30	37	Positive	38	2	12	9	5	2	8
		No effect	26	1	13	3	7	1	1
		Negative	3			2	1		
31–50	11	Positive	15	1	3	6		3	2
		No effect	6		3	2	1		
>50	7	Positive	5		2	2			1
		No effect	2		1				
No specified	3	Positive	3		3				

**Table 12 nutrients-14-01811-t012:** Recommended methodological guidelines for a future study on human physical performance of CoQ_10_.

Physical activity:	Four types: No physically active or light activity (<3 d/week), Moderately active (3–5 d/week), Very active (5–7 d/week), High level athletes.
Age (years):	Four types: ≤33, 34–48, 49–64, ≥65
Diet:	Homogenise the sample according to the type of diet
Dosage	4.0–4.5 mg/kg/d of Ubiquinol or Ubiquinone with Phytosome or Ubiquinone with vehicle to increase bioavailability, or explaining perfectly the diet related to the administration.
Placebo	Yes, double blinded.
Type of study	Parallel, never crossover to avoid training and supplementation effect
Tissue concentration (plasma)	Mandatory. Always after 24 h after last doses of placebo or study substance. Recommended before and after exercise stress test. Determined always as “μmol/mol Chol” and added as “μg/mL”.
Tissue concentration (muscle)	Very recommended. Indispensable to know mitochondrial density. Determined as “nmol/g protein”
Treatment period	>1 week.
Effort test before and after supplementation	Characterization of the maximal oxygen consumption of every subject by a graded maximal exercise test. Evaluation of the metabolic efficiency under, at and upper the anaerobic threshold. Long duration exercise at a submaximal level to evaluate oxidative and inflammatory pattern. Evaluation of subjective fatigue perception

## Data Availability

Not applicable.

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
