# Peer review of "Coenzyme Q10 Supplementation and Its Impact on Exercise and Sport Performance in Humans: A Recovery or a Performance-Enhancing Molecule?"

_nutrients, 2022, doi:10.3390/nu14091811_

Round 1
Reviewer 1 Report
Nutritional supplementation and its effect on exercise and sport performance is a topic of interest for both athletes and those seeking to improve their health and fitness. This is an informative, comprehensive, and detailed review on the subject. An extensive literature search was systematically conducted that includes both recent studies and previous related studies. The method for the search is described in detail. The tables are detailed and very informative. The table with recommended methodological guidelines for future studies on physical performance of CoQ10 is especially useful.
There are not an abnormal number of self-citations but a reference to “our study” on line 321 might be better worded.
In some cases the text is difficult to read with lengthy sentences such as on lines 201-204, 214-219, and 430-435.
The title “Coenzyme Q10 supplementation and its impact on exercise and sport performance in humans. A recover or a performing molecule?” might be better worded “Coenzyme Q10 supplementation and its impact on exercise and sport performance in humans: A recovery or a performance enhancing molecule?”
Author Response
Answer-Reviewer 1.-
There are not an abnormal number of self-citations but a reference to “our study” on line 321 might be better worded.
We appreciate the reviewer's recommendation. It is true that the way of expressing it is not very polite. We have taken the opportunity to restructure the whole paragraph in order to restate the ideas. We intend to give information that may be useful for the future, not that others have done it wrong, and certainly not that our opinion is the good or ideal one.
“However, it also warns us that an increase in plasma does not necessarily correlate with an increase in muscle tissue. It is possible that sufficient treatment time or higher dosage may be required, as it occurs in rodent models where it is shown that chronic ingestion of relatively large doses of CoQ10 in the diet is able to increase the CoQ10 concentrations especially in the mitochondrial fractions of heart in (114). It is therefore worth remembering that the use of the same dose of Ubiquinone, or Ubiquinol, taken orally by athletes with different body composition, leads to different intakes of CoQ10 per kg of body weight (33). On the other side, the change in muscle and plasma levels in relation to the CoQ10 administration is inconsistent. Thus, Svensson et al (69), Zhou et al (79), Cooke et al (46) and Laaksonen et al. (59) in the oldest athletes, show an increase in plasma that was not reflected in muscle. In contrast, Laaksonen et al (59) in the young athletes and Drobnic et al (48) do observe an increase in concentration, although not significant, before and after supplementation. This discrepancy may be due, in addition to the CoQ10 regimen administered, to another methodological aspect to mention. Muscle biopsy is usually performed by needle biopsy. Although fine needle aspiration is a well-referenced model, the samples are relatively small (15-30 mg Cooke, 40-80 mg Svensson, 50-100 mg Zhou), which are, in our case intended to illustrate the entire composition of the muscle mass of a high level or veteran athlete. This type of samples is undoubtedly excellent for discriminating and diagnose pathology but limit the repeatability of assessments when quantifying molecules, so a larger muscle sample is much appreciated by the laboratory pathologist. In the last study discussed (48, 99), a muscle biopsy sample of 200-250 mg was obtained from the vastus lateralis muscle using the ENCOR ULTRA® 10 G biopsy system under ultrasound guidance. This is a simple and practical method, although not cheap, but it provides a high quality homogeneous sample, very suitable for the study of muscle tissue, and allows portions to be retained for various subsequent analyses and pooling if a multicentre project is considered.”
In some cases the text is difficult to read with lengthy sentences such as on lines 201-204, 214-219, and 430-435.
201-204
The inclusion criteria administered in the review are indeed strict. It is true that some of them could be omitted, e.g. that the study can be cross-over, that studies with a control group and not only placebo are included, or that it is not determined whether or not the tested molecule increases in tissues. If one focuses only on the objective and considers a correct methodological structure for assessing the effect on performance, there are many deselected studies that certainly deserve to be considered. For this reason, tables 3a to 5b include the rest of the studies screened and some of them are discussed.
430-435
Individuals with higher mitochondrial density enjoy a higher margin of functional mitochondrial capacity, which perhaps explains why the effect of COQ10 when administered to them is greater. It should be remembered that mitochondrial capacity may be underexpressed in individuals who, although not physically active at the start of the studies, were very active in their youth and have a pool of latent mitochondria that have not disappeared (167). This is also the case of the athletes who have taken anabolic steroids (168). This increased mitochondrial responsiveness can be activated with an appropriate physical and nutritional stimulus. It is quite possible that the change in performance would be different in individuals who never played sport intensively in their youth tan those who did. In any case, the results would be homogenised across studies if mitochondrial density were determined.
The title “Coenzyme Q10 supplementation and its impact on exercise and sport performance in humans. A recover or a performing molecule?” might be better worded “Coenzyme Q10 supplementation and its impact on exercise and sport performance in humans: A recovery or a performance enhancing molecule?”
The comment is very appropiate. We have changed the Title. Thank you
Reviewer 2 Report
The authors reviewed the impact of CoQ10 supplementation on exercise and sports performance. They selected 59 articles from the literature databases, and nine articles were included in the systematic review. Although the data are inconclusive due to a great diversity of works with different orientations, they reasoned that CoQ10 might function as a recuperative rather than an ergogenic substance in humans. Finally, they proposed a methodological guideline for future studies on the physical performances of CoQ10 to facilitate more precise analyses.
The reviewer thinks that the review might be helpful as a database collecting studies on the beneficial effect of CoQ10 on exercise and sports performance, although the results were still inconclusive. The reviewer suggests the following for improvement of the manuscript.
- Line 109, dilucidate can be changed to elucidate. Obsolete words should be replaced with contemporary words.
- Figure 1, excluded articles should be shown in the side column (left or right side). Additionally, the description of eligibility could be moved to the legend of Figure 1.
- Tables 3a to 5b can be removed and shown as supplemental materials, as they are a kind of database of excluded papers from the systematic review.
- Lines 394, 398, and 402, VO2max should be VO2max like line 405.
- Lines 401-402, “On the other ide” might be a misspelling of “On the other side.”
6. Lines 456-464 and table 8, the reviewer hardly follows how the authors made a methodological guideline after reviewing. Please describe the grounds for how to set up each requirement.
Author Response
Answer-Reviewer 2.-
Line 109, dilucidate can be changed to elucidate. Obsolete words should be replaced with contemporary words.
Word modified
Figure 1, excluded articles should be shown in the side column (left or right side). Additionally, the description of eligibility could be moved to the legend of Figure 1.
Modified
Tables 3a to 5b can be removed and shown as supplemental materials, as they are a kind of database of excluded papers from the systematic review.
We strongly agree with the reviewer that the data in the mentioned tables do not correspond to the eligibility of the systematic review. However, due to the strict conditions of eligibility for this review, we consider that very interesting research has been done on this molecule, some of the results of which are discussed in the discussion. If it does not seem too bad to the reviewer and in order to facilitate the reading and evaluation of data from all research on CoQ10, in humans and in relation to exercise, we keep the tables in the main document. We recognise that moving these tables to an Annex limits this evaluation and generation of a proper criterion, especially with respect to the proposed idea of generating a more homogeneous research model.
Lines 394, 398, and 402, VO2max should be VO2max like line 405.
Changed
Lines 401-402, “On the other ide” might be a misspelling of “On the other side.”
Modified
- Lines 456-464 and table 8, the reviewer hardly follows how the authors made a methodological guideline after reviewing. Please describe the grounds for how to set up each requirement.
We have added these comments and 7 references.
There are four types of populations that are markedly different in the possible action of CoQ10 in relation to physical performance: the sedentary, the moderately physically active and the very active (186,187). which in turn correspond to different age groups. Functionality, learning, memory and mitochondrial density are likely to behave differently after exposure to the supplement. The proposed table shows four age ranges, which are somewhat arbitrary and could be adjusted to human age periods (188), but since these correspond more to concepts of mental maturity and senility, it is considered more interesting to associate them with aspects of physical maturation and exposure to pathology (189) rather than purely psychological or mental aspects.
The administered dose of both the oxidised and reduced forms must be adequate to ensure that bioavailability is correct. A dose is proposed which, based on the latest studies with Ubiquinol, shows an adequate increase in plasma levels. Moreover, the period of administration must be given a minimum of time to justify that the functional results correspond to the presence of the element studied in the place where it functions, the mitocondria (102,109). This thinking justifies the determination of levels not only in plasma, which we consider mandatory, but also recommends determination in muscle tissue to associate it with the mitochondrial characteristics of interest to us. In addition, it is recommended that at least one biopsy be taken from the vastus lateralis quadriceps if it is decided to do several. The different concentration in humans is not fully defined but it is presumable that it occurs as in rodents and its concentration is different depending on the muscle examined (189,190). In turn, plasma determination should not be done when the treatment has been administered at a time when plasma availability is high due to absorption. It is therefore justified that at least 24 h should elapse, taking into account the bioavailability of the substance (16).
It is also necessary to take into account other aspects related to the idiosyncrasies of the subject in relation to the group studied, such as the diet maintained, excessive exercise, or the sedentary lifestyle of adults who did high-intensity sport in their youth, as explained in the discussion.
184.- Burke LM, Peeling P. Methodologies for Investigating Performance Changes With Supplement Use. Int J Sport Nutr Exerc Metab. 2018; 28(2):159-169. doi: 10.1123/ijsnem.2017-0325.
185.- Grossman J, Mackenzie FJ. The randomized controlled trial: gold standard, or merely standard? Perspect Biol Med. 2005 Autumn;48(4):516-34. doi: 10.1353/pbm.2005.0092.
186.- WHO Guidelines on Physical Activity and Sedentary Behaviour. Geneva: World Health Organization; 2020. PMID: 33369898. Bookshelf ID: NBK566045
187.- ACSM's Guidelines for Exercise Testing and Prescription. American College of Sports Medicine Filadelfia: Wolters Kluwer Ed. Riebe, D, Ehrman, JK, Liguori, G, Magal M. (2018).
188.- Geifman N, Cohen R, Rubin E. Redefining meaningful age groups in the context of disease. Age (Dordr). 2013; 35(6):2357-66. doi: 10.1007/s11357-013-9510-6
189.- Beyer RE, Morales-Corral PG, Ramp BJ, Kreitman KR, Falzon MJ, Rhee SY, Kuhn TW, Stein M, Rosenwasser MJ, Cartwright KJ. Elevation of tissue coenzyme Q (ubiquinone) and cytochrome c concentrations by endurance exercise in the rat. Arch Biochem Biophys. 1984; 1;234(2):323-9. doi: 10.1016/0003-9861(84)90277-7.
190.- Gohil K, Rothfuss L, Lang J, Packer L. Effect of exercise training on tissue vitamin E and ubiquinone content. J Appl Physiol (1985). 1987; 63(4):1638-41. doi: 10.1152/jappl.1987.63.4.1638.

This manuscript is a resubmission of an earlier submission. The following is a list of the peer review reports and author responses from that submission.
Round 1
Reviewer 1 Report
Thank you for your work on an interesting review on such a widely touted supplement. Attempting to review papers considering CoQ10 as an adjuvant medical treatment and as an ergogenic aid for sports performance is a difficult task. Given the recent reviews of CoQ10 use in clinical settings, it would behoove the authors to perhaps limit this review to the sports performance side. As written the potential of performance changes from CoQ10 (or derivatives) seemed to be where the majority of the information included was targeted anyway. I believe doing this will help reduce the amount of information needing to be conveyed and thus assist in shortening the paper and its tables, which is always a challenge in a review paper.
There did seem to be a lot of information and background in the introduction that was only tangentially related to the topic at hand. Too many terms were defined and explained. Again, deciding what to assume the reader will understand without such explanations is always difficult. I believe focusing on just the performance side of the issue will help with this as well. Even with that decision, I would consider removing or significantly reducing the information provided in paragraphs 2, 3, and 4, while reducing the verbosity of paragraphs 5-8. I might even consider leading the introduction, rather than finishing it, with the information in paragraph 8.
The methods also could be reduced somewhat but more through re-wording than eliminating the information provided, which was pertinent for the reader. I do see the utility of the table in this section (I did not see a title), but thought having the information there might allow for less information to be provided in-text.
As in the section above, I felt the results could be streamlined. This would come by default if the review did focus on the sport-performance issue as suggested above. Throughout the text in this section, the focus seemed to be more on an analysis of the quality of the studies than on the actual findings. As both are important, perhaps there is a better balance to be stuck. More importantly, I strongly encourage the authors to consider alternate arrangements of the information conveyed in tables 1-4. Perhaps ordering the studies within these tables by their findings/results as indicated in the column headings: Exercise Testing (eliminating the non-exercise studies would automatically resolve that) then, any of the Measured Parameters and Effect of CoQ10 (Antioxidant Pattern, Physical Performance, etc.). Doing so would make it much easier for the reader to recognize patterns and be aware of repeated findings across studies. It may make sense to simply include these as a figure for the appendix and use the summary tables (5-7) in the article. I found these summary tables to be quite useful, but could also be rearranged similarly as suggested above, as opposed to alphabetically. Again, these tables were very helpful as written, but depending on how the other tables are amended, the supplementary tables could become superfluous.
In the first paragraph of the discussion, the 2 recent reviews on CoQ10’s clinical use are cited and the paper goes on to state that what is undetermined surrounds the ergogenic aid benefit to CoQ10’s use in sports. That being the case, this review should do exactly that, and target this issue specifically. As such, I would suggest eliminating or reducing the upper 2 discussion paragraph along with sections 4.1 and 4.2 entirely. The remainder of the paper would remain pertinent, but it, as well as the other sections of the manuscript still require significant editing from a structural standpoint. Specific word choices and phrasings did not always seem to capture what I believe was intended. Without getting bogged down in the specifics, an example would be the use of the phrase “For some reason” to start the second paragraph of section 4.3 (line 355). This is a less formal/more familiar phrase to convey that the reason is unknown. The paper goes on to give the reason. Such occurrences can be found throughout the paper. Further editing of such items would elevate the level of this paper, which was clearly written with a great deal of thought and effort from the authors.
Reviewer 2 Report
The use of supplements and their effects in performance is always a very interesting topic and deserves a special attention. This paper addresses one of the most promising supplement in terms of well-being, health promotion and that could also be an ergogenic in special conditions. The review covers the most relevant studies on the subject and is just lacking a better background and methodology writing.
Abstract
I would suggest you use a abstract structure that includes 1 background sentence, the aim of the study , the research method and some results about studies found, samples features, the sports, the supplement/substance used and an overall conclusion from these results.
Introduction
You are missing some more relevant references on what is an ergogenic substance and you can also refer to cognitive function ergogenics. There are very few reference backgrounds in introduction.
Considering that it is not a systematic review, you could focus more your introduction on CoQ10 itself
Methods
I suggest you divide in subtopics/subsections such as:
Databases
Keywords used
Inclusion and exclusion criteria
The table in page 4 is not referred in the text.
Results
I suggest you do a paragraph (or more) to systematize the results referring: how much studies, the sample sizes interval etc.
Discussion
Some references are lacking to sustain the text like in line 355-358
Reviewer 3 Report
Comments to the Author:
I thank to the editors for the opportunity to review this study, beside I would also like to congratulate the authors for the made effort in their study. The present manuscript by Drobnic et al., analyzed “Coenzyme Q10 supplementation and its impact on exercise and sport performance in humans. A recover or a performing molecule?”. The authors with this review try to provide a review of most of the studies that correspond to the evaluation of CoQ10 with sports and physical performance, aiming to provide an orientation as to whether its function can be considered as a particularly recover or ergogenic molecule in itself. The main problem with your review is that authors do not really express with precision what you have done. This is because the authors showed the structure of a systematic review, however they do not follow such a methodology and show a simple narrative review. Therefore, I recommend the authors to make an effort and adapt their review to a systematic review, following all the necessary methodological processes.
General Comments:
- The review has many unreferenced arguments, mainly in the introduction. Furthermore, until paragraph 77 the authors say absolutely nothing about the main topic of the review, which is performance, recovery and CQ10 supplementation, basically introducing irrelevant information.
- The authors should add more information about relationship between Q10 supplementation and its effect in sport performance and recovery process. Particularly, authors must improve this section (introduction) and explain what type of physiological pathways are interacting.
- I am pleased to know that your systematic search has been carried out in the “Pubmed, and Web of Science” database, since they are ones of the most important databases. However, I don't understand why you used more databases than these. Could you give me any good reasons?
- I note in your methodology section that you use different criteria to evaluate the scientific evidence of the studies analyzed, another indication that you intend to conduct a systematic review rather than a review. Therefore, I recommend to the authors not to make a mix between a review and a systematic review, but to focus on a systematic review.
- The first paragraph of the discussion should be much better structured. First the main aim of the study and then the most relevant results. It is not necessary to add more information because it confuses the reader.
- In the discussion section there is a session on antioxidant activity, Could the authors argue to me how this paragraph relates to sports performance, recovery and CQ10 supplementation? In addition, there is another section on muscle injuries and inflammatory processes where the authors only dedicate a small part to the effects of CQ10 on these two processes. Authors should be aware of the significance of the discussion section. There is too much information in this section without any relevance, just general information about muscle damage, antioxidant activity, etc. Try to carry out a real discussion between the results you have obtained and the existing literature. There is no real discussion in their study.
- What were the strengths and limitations of this review?